# Impact of Heart Disease Risk Factors, Respiratory Illness, Mastery, and Quality of Life on the Health Status of Individuals Living Near a Major Railyard in Southern California

**DOI:** 10.3390/ijerph15122765

**Published:** 2018-12-06

**Authors:** Kelly Baek, Semran K. Mann, Qais Alemi, Akinchita Kumar, Penny Newman, Rhonda Spencer-Hwang, Susanne Montgomery

**Affiliations:** 1Department of Social Work & Social Ecology, School of Behavioral Health, Loma Linda University, 1898 Business Center Drive, San Bernardino, CA 92408, USA; skmann@llu.edu (S.K.M.); qalemi@llu.edu (Q.A.); akinchita@gmail.com (A.K.); smontgomery@llu.edu (S.M.); 2Formerly Affiliated with the Center for Community Action and Environmental Justice, P.O. Box 33124, Jurupa Valley, CA 92519, USA; pennynewman20@gmail.com; 3School of Public Health, Loma Linda University, 24951 North Circle Drive, Loma Linda, CA 92350, USA; rspencer@llu.edu

**Keywords:** health status, heart disease risk factor, low-income, mastery, quality of life, railyard, respiratory illness

## Abstract

The potential health risks for communities that surround railyards have largely been understudied. Mastery and quality of life (QoL) have been associated with self-reported health status in the general population, but few studies have explored this variable among highly vulnerable low-income groups exposed to harmful air pollutants. This study investigates the relationship between self-reported health status and correlates of Heart Disease Risk Factors (HDRF) and Respiratory Illness (RI) with mastery and QoL acting as potential protective buffers. This cross-sectional study of 684 residents residing near a Southern California railyard attempts to address this limitation. Results from three separate hierarchal linear regressions showed that those who reported being diagnosed with at least one type of HDRF and/or RI reported lower perceived health status. For those that lived further from the railyard, mastery and QoL predicted modest increases in perceived health status. Results suggest that mastery and QoL may be helpful as tools in developing interventions but should not solely be used to assess risk and health outcomes as perceived health status may not measure actual health status.

## 1. Introduction

Major transportation hubs are notorious for contributing to poor air quality, namely airborne pollutants from diesel exhaust such as particulate matter (PM) and nitrogen oxides (NO_x_), which are known carcinogens, according to the Environmental Protection Agency [1]. Studies show that health impacts on communities in close proximity to major transportation hubs, such as railyards, are not limited in their impact to respiratory diseases such as asthma [2,3] chronic obstructive pulmonary disorder (COPD) [4], and lung cancer [5], but also increases the risk of premature death [5,6], cancer [7] and heart disease [8].

In addition, studies suggest that the risk of developing heart disease significantly increases for those who live near railyards due to little to no physical activity and unhealthy diets due to the lack of safe places to exercise and access to fresh foods, respectively, within low income communities [9,10]. The American Heart Association concluded that exposure to PM air pollution also contributes to cardiovascular morbidity and mortality [11]. Other studies have associated fine PM with a variety of respiratory and cardiovascular problems, which include but are not limited to increased hospitalizations for cardio-respiratory causes, aggravated asthma, other respiratory symptoms, acute bronchitis, irregular heartbeat, heart attacks, and premature death in people with heart or lung diseases [12]. Exposure to PM that are smaller than 2.5 μm for only a few hours to a week can lead to cardiovascular disease related mortality and non-fatal events. In comparison, long-term exposure increases the risk for cardiovascular mortality to a greater extent, which leads to reductions in life expectancy by several months to a few years [11]. Furthermore, Brooks et al.’s studies [11,13] found that living near a railyard greatly increases one’s exposure to air and noise pollutants. Environmental noise, especially noise that is caused by transportation, can cause sleep disturbances [14]. Poor sleep, often a side effect to noise pollution, is linked to a number of long-term cardiometabolic, psychiatric, and negative social outcomes [14] which can cause additional stress on the body leading to detrimental effects on cardiovascular health [11].

### 1.1. San Bernardino Railyard

In Southern California, the topography of the San Bernardino Mountains form a natural barrier that allows air pollutants brought in from the Los Angeles basin by prevailing winds, to be trapped [15]. In addition, the area has a number of its own local air pollination sources from the dense population and local industry and numerous freeways and highways, that with its topographical and meteorological characteristics, it results in some of the poorest air quality measured in the US [12]. One of these local polluters is the San Bernardino Railyard (SBR) facility. Based on risk assessments conducted by the California Air Resources Board (CARB), the SBR ranks among the top five most polluting railyards in California, and first in terms of community health risk due to the large population living in the immediate vicinity [16].

Low-income communities are often located near industrial railyards as a result of limited affordable housing options elsewhere. Since these low-income populations already often face poorer health outcomes, their increased exposure to air pollutants from the railyards only exacerbates that vulnerability [11,13]. Likely, health-threatening effects of poor air quality are further compounded by the significant psychological stress of day-to-day survival that accompanies poverty [17]. Indeed, the population living next to the SBR is predominantly young (including a large proportion of children), low income, and largely Latino. Available health outcomes data suggest tremendous health disparities between the region’s African American, Latino, and Caucasian population [12]. While the overall county’s poverty rate is 15.8%, the poverty rate for Latinos stands at 34.9%, which far exceeds the overall poverty rate for the state (14.2%), the nation (12.4%) and California’s overall Latino poverty rate of 28% [18]. Further limiting available support for community members was the city of San Bernardino’s bankruptcy in 2012, making this region one of southern California’s poorest municipalities, with a disproportionate number of neighborhoods facing a host of socio-economic, health, and environmental challenges.

### 1.2. The ENRRICH Project

Faced with these challenges, community members voiced an urgent call to action to the City’s Mayor, lawmakers, and local researchers to address these environmental concerns. In collaboration with residents and the Center for Community Action and Environmental Justice (CCAEJ), a local community-based organization, researchers from Loma Linda University (LLU) responded by forming the Environmental Railyard Research Impacting Community Health (ENRRICH) project. Using a community-based-participatory-research (CBPR) approach, ENRRICH aimed to explore the health risks of residents living in close proximity to the SBR and to support the development of a community response plan by emphasizing the significant role of community input, ownership, and concerted actions to produce appropriate, innovative and practical solutions that are cost-effective and sustainable. Field teams comprised of community workers from CCAEJ, in partnership with LLU, also went door to door in the community to collect noninvasive biological tests and survey. In addition, the researchers created three zone distances based on the prevalence of adverse health effects among exposed adults by assessing the spatial gradient of air pollution and associated health risks using computer-based modeling to estimate the transport and dispersion of diesel emissions from the railyard. Zone 1 was defined as less than a mile from the railyard, Zone 2 as 1–3 miles, and Zone 3 as 3–5 miles from SBR. While all three zones exhibited higher air pollution levels than the average for the surrounding area, residents in Zone 1 were identified as being exposed to significantly higher concentrations of air and noise pollution than the other two zones.

While air pollution is a concern for the community, it is not the only one. Findings from a qualitative needs assessment suggest [13] that while community members were concerned about the poor air quality and related health risks, other challenges to them had even higher priority. The findings indicated that members residing near the railyard faced a number of obstacles to their QoL that stemmed from a high level of community violence, economic problems, homelessness, railyard-related noise exposure, and lack of access to healthcare, especially for their children, many of whom suffer from respiratory conditions.

Despite the many obstacles that low-income communities residing near railyards face, studies of the general population have identified the potential positive impact of QoL and mastery on self-reported health for those diagnosed with heart disease [19,20,21] or respiratory illness [19,22]. According to the World Health Organization, QoL is defined as:

An individual’s perception of their position in life in the context of culture and value systems in which they live and in relation to their goals, expectations, standards, and concerns. It is a broad ranging concept affected in a complex way by the person’s physical health, psychological state, personal beliefs, social relationships, and their relationship to salient features in their environment [23].

Mastery refers to the extent an individual feels in control over events in their lives [24]. Specifically, higher perceived QoL and greater sense of control over their treatment were linked to higher perceived health status for those that had been diagnosed with COPD [22] or with cardiovascular disease [20,21] suggesting that further studies be done that explores the impact of adaptable personal characteristics and perceptions as potential protective buffers for health status.

The current study assesses the potential impact of self-reported heart disease risk factors (HDRF) and respiratory illness (RI) diagnosis on perceived health status. The authors explored if (1) if distance from the railyard influenced perceived health, (2) HDRF and RI significantly impacted health status, and (3) perceived mastery and QoL served as protective buffers against self-reported HDRF and RI diagnosis, in addition to contributing to better-perceived health in this community. Specifically, we hypothesized that (1) those that lived closest to the railyard would report more adverse health status, (2) those diagnosed with HDRF and RI would report significantly lower health status, (3) those with higher perceived mastery and QoL would report better perceived health, and (4) that higher perceived mastery and QoL would decrease the impact of HDRF and RI on health status.

## 2. Materials and Methods

The data for this paper was collected as part of the aforementioned Project ENRRICH study, which was carried-out in two consecutive cross-sectional waves (during summer of 2011 and winter/spring of 2012) in order to account for seasonal variation in air quality. A random household sample of 684 residents was collected to ensure a broad representation. The sampling strategy was based on distance from the railyard. The zones were defined across the spatial gradient of air pollution and associated health risks based on the 2008 California Air Resources Board (CARB) Health Risk Assessment (HRA) report [24] that was originally derived through the implementation of computer-based modeling used for estimating the transport and dispersion of diesel emissions from the railyard. Zone distances were selected based on the prevalence of adverse health effects among exposed adults. Zone 1 was within one mile of the railyard, Zone 2 between one and three miles from the railyard, and Zone 3 between three and five miles from the railyard. Every household in Zone 1 was sampled to match the sharp decline of diesel particulate matter concentrations, which was postulated to occur over a short distance from the railyard. Sampling in zones 2 and 3 employed a t-stage clustering sampling methodology in which sixty census blocks within a sampling zone were first randomly selected and then a set of five houses within each Census block was chosen. Selected houses were randomly selected using a GIS-based random number generator. Interviews lasted approximately 60–90 min and were conducted in English and Spanish based on respondent preference by trained research assistants. Participants signed consent forms that were approved by the university’s Institutional Review Board. Non-invasive biological tests were conducted (respiratory tests such as using NIOX MINO devices to collect airway inflammation data via fractional exhaled nitric oxide) and a survey to collect information on demographics, exposures, health status, lifestyle, self and community perceptions.

### 2.1. Measures

#### 2.1.1. Dependent Variable

*Health Status*. Perceived health status was comprised of one question. Respondents reported their health from a range of 1 (poor) to 4 (excellent) with higher scores representing better-perceived health.

#### 2.1.2. Independent Variables

Our main independent variables of interest were related to respondents reporting having a diagnosis of either heart disease risk factors (HDRI) and/or respiratory illness (RI). While the survey asked about other HDRF and RI, the most commonly reported HDRI and RI that were above the state and national benchmark were selected to create the most parsimonious model.

*Heart Disease Risk Factors (HDRF).* Respondents were asked if a physician had ever informed them of having angina and/or high blood pressure using a binary yes (coded ‘1’)-no (coded ‘0’) response choice.

*Respiratory Illness (RI).* Respondents were also asked if a physician had informed them of having asthma, COPD or any bronchial condition, again in a binary yes-no response choice format.

#### 2.1.3. Moderating Variables

*Quality of Life.* The QoL scale was guided by qualitative study that asked community members living near the railyard about their perceived QoL and health challenges by Spencer et al. [25]. The scale was composed of 22 questions, including ‘I feel safe walking in my community, day or night’, ‘people in my community generally get along with each other’, ‘Local facilities in my community offer many opportunities to get exercise’, ‘The fresh fruits and vegetables available in my community are of high quality’ and ‘The noise from my community keeps me awake or wakes me up in the middle of the night’ and were based on a Likert scale ranging from ‘1’ (“strongly disagree”) to ‘5’ (“strongly agree”). Several items were reverse coded to accommodate the overall directionality of the scale with higher values representing higher QoL. All the scores were added up and then averaged back to the original scale. Scores range from 1 to 5 with higher scores indicative of higher perceived QoL. The scale demonstrated adequate internal reliability (Cronbach’s α = 0.74).

*Mastery*. A single item from the Pearlin Mastery Scale [26] was used to measure perceived mastery. Respondents were asked to what level they agreed with the statement ‘I can do just about anything that I set my mind to’ ranging from 1 (‘strongly disagree’) to 5 (‘strongly agree’). Higher scores represented higher levels of mastery.

#### 2.1.4. Control Variables

*Demographic Variables*. We assessed respondents’ age, gender, race/ethnicity (White, African-American, Hispanic or Latino, or Other), level of education (‘high school or less’ or ’some college and more’), marital status (not married/married), and average household income (‘less than $30,000/more than $30,000’).

*Health Insurance.* One question assessed if and what type of health insurance the respondents had. This variable was coded as a dichotomous variable indicating if they did not have (0) or have health insurance (1).

*Health Care Access*. One question asked if there was a place that the respondents went to when sick or in need of advice about their health. If they replied yes, they were probed on where they went, for example a physician’s office, clinic, county public health department clinic, or other specified locations. Those who did not have access or stated that they went to the emergency room were categorized as not having regular health care access. The responses were collapsed and then dichotomized with ‘0’ indicating that they did not have access to regular care and ‘1’ indicating that they did have access to regular care.

*Functional Impairment*. Perceived functional impairment was a single item indicator that asked how much their health hindered them from enjoying life, with response choices ranging from 1 (‘daily’) to 5 (‘none at all’). Higher scores indicated that respondents are less functionally impaired.

## 3. Results

### 3.1. Descriptive Analysis

Table 1 displays the socio-demographic characteristics of our participants. Two thirds of the 684 respondents identified as Latino (64.6%), whereas 12.9% identified as White, 10.4% as African American, and 12.1% as Other. Half of those who identified as Latino were monolingual. 

The sample had relatively low levels of education and income, with 63.2% having a high school education or less and over 66% of the respondents reporting that they made less than $30,000 a year. In addition, over 42% of the respondents were unemployed and most were below the California poverty levels for household income. Please refer to Table 1 for complete demographic information.

Bivariate tests were conducted to examine the socio-demographic variables, how much their health hindered from enjoying life, the presence of heart disease correlates, respiratory illness diagnoses, as well as mastery and QoL for each zone. Overall, we found that perceived health status did not significantly differ by gender, income, or if they had access to regular care. However, for respondents that lived closest to the railyard (Zone 1), those that identified as Latino, were married, had less than a high school education, were older, had been diagnosed with angina and/or high blood pressure, felt more impaired by their health, felt less mastery over their lives, and had lower perceived QoL scores reported significantly lower perceived health.

For respondents that lived in Zone 2 (1–3 miles from the railyard), those that identified as White, did not have health insurance, were younger, had not been diagnosed with any type of heart disease risk factor or respiratory illness, did not feel that their health impaired their functioning, felt they had greater control over events in their lives, and had higher perceived QoL scores reported significantly higher perceived health status.

Latinos reported significantly lower perceived health status in contrast to Whites who reported higher perceived health for Zone 3 (live 3–5 miles from the railyard). Respondents who were more educated, had not been diagnosed with either a diagnosed heart disease risk factor or respiratory illness, felt that their health did impair daily functioning, and had perceived QoL also reported higher health status. Overall, heart disease risk factors, functional impairment, and QoL significantly impacted health status across all three zones. Please refer to Table 2 for the relationship between the categorical demographic variables and health status and Table 3, Table 4 and Table 5, for the correlations between the continuous variables and health status. 

### 3.2. Multivariate Analysis

Three hierarchical linear regression analyses, one for each zone, were run. The first step controlled for socio-demographic variables significantly associated with perceived health status at the bivariate level (please refer to Table 2, Table 3, Table 4 and Table 5), the second step determined the influence of HDRF and RI and the last step assessed the amount of variance of perceived mastery and QoL on perceived health status. The interactions between the illness variables (HDRF and RI) and the protective buffers (mastery and QoL) were omitted from the final model due to Variance Inflation Factor scores being outside of the acceptable range of −2 to 2 (indicating that there is multicollinearity). All other interactions between the variables were within the normal range. The assumptions of normality and homoscedasticity were assessed via scatterplots. Neither of these assumptions was violated.

For Zone 1, the variables in Step 1 significantly contributed to predicting health status (*R^2^* = 0.13, *R^2^_adjusted_* = 0.11, *F*(5, 194) = 2.95, *p* < 0.001), with ethnicity (*β* = 0.17, *p* < 0.05) and functional impairment (*β* = 0.30, *p* < 0.001) being statistically significant. Step 2 showed that after controlling for socio-demographic variables, HDRF (*β* = −0.26, *p* < 0.001) contributed an additional 6% variance to the model (*R^2^* = 0.19, *R^2^_adjusted_* = 0.16, *F*(6, 192) = 7.21, *p* < 0.001), supporting the hypothesis that self-reported HDRF diagnosis would negatively impact self-reported health status. However, mastery nor QoL significantly contributed to the model, hence failing to support the hypothesis that mastery and QoL would have a positive impact on health status or decrease the effects of HDRF. Ethnicity (*β* = 0.16, *p* < 0.05), functional impairment (*β* = 0.20, *p* < 0.01), and HDRF (*β* = −0.25, *p* < 0.01) remained significant. Overall, those that identified as Latino, reported that their health interfered with life functioning, and had been diagnosed with HDRF were significantly more likely to report poorer health. In addition, it did not appear that mastery or QoL served as a potential protective factor against HDRF. See Table 6 below.

For Zone 2, the first step also significantly predicted health status (*R^2^* = 0.30, *R^2^_adjusted_* = 0.28, *F*(4, 214) = 21.77, *p* < 0.001), with ethnicity (*β* = −0.13, *p* < 0.05), health insurance (*β* = −0.14, *p* < 0.05) and functional impairment (*β* = 0.49, *p* < 0.001) being statistically significant. Step 2 showed that after controlling for socio-demographic variables (Step 1), RI (*β* = −0.12, *p* < 0.05) contributed an additional 2% variance to the model (*R^2^* = 0.32, *R^2^_adjusted_* = 0.30, *F*(6, 212) =10.75, *p* < 0.001), supporting the hypothesis that self-reported RI diagnosis would negatively impact self-reported health status. The variables ethnicity (*β* = −0.14, *p* < 0.05), health insurance (*β* = −0.14, *p* < 0.05), and functional impairment (*β* = 0.42, *p* < 0.001) remained significant. In step 3, mastery (*β* = 0.14 *p* < 0.05) and QoL (*β* = 0.12, *p* < 0.05) contributed 3% of the variance to the model, supporting the hypothesis that mastery and QoL would have a positive impact on health status (*R^2^* = 0.35, *R^2^_adjusted_* = 0.31, *F*(8, 210) = 9.83, *p* < 0.001). Ethnicity (*β* = −0.15, *p* < 0.05), health insurance (*β* = −0.14, *p* < 0.05), and functional impairment (*β* = 0.36, *p* < 0.001) remained significant. 

Overall, respondents that identified as white, had insurance, felt that their health did not hinder their daily functioning, had not been diagnosed with RI were more likely to report higher perceived health status. In addition, when mastery and QoL were added to the model, RI no longer remained significant supporting the hypothesis that mastery and QoL would decrease the impact of RI on health status. See Table 7 above.

Step 1 significantly predicted health status (*R^2^* = 0.13, *R^2^_adjusted_* = 0.25, *F*(5, 235) = 15.01, *p* < 0.001), for Zone 3 with functional impairment (*β* = 0.46, *p* < 0.001) significantly contributing to the model. Step 2 showed that after controlling for socio-demographic variables (Step 1), RI (*β* = −0.14, *p* < 0.05) contributed an additional 4% variance to the model (*R^2^* = 0.29, *R^2^_adjusted_* = 0.26, *F*(6, 233) = 13.06, *p* < 0.001), supporting the hypothesis that self-reported RI diagnosis would negatively impact self-reported health status. The variables ethnicity (*β* = 0.15, *p* < 0.05) and functional impairment (*β* = 0.40, *p* < 0.001) were also significant. However, mastery nor QoL significantly contributed to the model, hence failing to support the hypothesis that mastery and QoL would have a positive impact on health status. Ethnicity (*β* = 0.16, *p* < 0.05), functional impairment (*β* = 0.20, *p* < 0.01), and RI (*β* = −0.14, *p* < 0.05) remained significant. Overall, those that identified as non-White, reported that their health interfered with life functioning, and had been diagnosed with a respiratory illness were significantly more likely to report poorer health. In addition, it did not appear that mastery or QoL served as a potential protective factor against RI. See Table 8 below.

## 4. Discussion

This major focus of this study was to investigate the relationship between self-reported health status and heart disease risk factors, respiratory illness, and potential protective factors. The communities that surround the San Bernardino Railyard are very low-income and face poorer health outcomes and increased exposure to air pollutants [11,13]. We believe that this increased exposure to poor air quality increases the psychological stress these minority families face in their day-to-day survival [17], which was supported by the results. We found that mastery and QoL did not serve as protective buffers for HDRF and RI for respondents that lived in Zone 1. However, for respondents that lived in Zone 2, after mastery and QoL was added to the model, RI no longer significantly contributed to the model suggesting that sense of control over one’s life and QoL could serve as protective buffers for health status for residents that lived further from the railyard. This contrast also served to emphasize the vulnerability of the families that live in the closest proximity to the railyard compared to respondents that lived at least a mile away from the railyard.

While mastery, QoL, and respiratory health were not related to perceived health status for those that lived closest to the railyard, perceived functional impairment and heart disease risk factors had a significantly negative influence on respondents’ perceived health status. This finding is supported by studies that indicated a feeling that they have less control over events in their lives has detrimental effects on one’s health, leading to higher rates of illness and slower recovery times [19,20,21,22]. In contrast, higher perceived mastery and QoL appeared to have a positive relationship with health status at both the bivariate and multivariate level for respondents that lived in 1–3 miles from the railyard. Even within this very low income area that has struggles with poverty and crime, greater sense of control over their lives and more positive perceptions of their QoL, predict better-perceived health status. Respondents that lived furthest from the railyard did not appear to be as adversely impacted by HDRF and RI (even though mastery and QoL did not appear to serve as a protective factors) further supporting that physical distance from the railyard does have an impact on health status.

The significant differences in health status among ethnic groups, with Latinos reporting significantly lower health status than Caucasians regardless of income, aligns with studies that have investigated neighborhood socio-economic status (SES) and related obesity risk. These studies had similar findings, in that Mexican-Americans were impacted more by lower neighborhood SES with respect to prevalence of overweight (higher body mass index) and obesity [27,28].

The bivariate relationship between health status and whether one had health insurance and/or access to regular health care appeared to be contradictory in which those without health insurance and access to regular care were more likely to report higher health status. We hypothesize that in this low income environment the cost of health insurance is a luxury only obtained either by participants who had poorer health or that alternatively, those who do not have insurance may not know that they have a diagnosable condition such as HDRF and RI. Therefore, further analyses were conducted to assess the relationship between those that had been diagnosed with either RI or HDRF and if they had insurance and/or access to regular care. A one-way ANOVA revealed that those who had access to regular care or had health insurance and had been diagnosed were more likely to report poorer health compared to those had been diagnosed but that did not have access to regular care/insurance, had insurance/access with no diagnosis, and no access/care and no diagnosis.

### Limitations

Given the location-specific nature of the ENRRICH study, there are some noteworthy limitations that should be taken into consideration. Foremost, the ENRRICH Study intentionally sampled a specific population of individuals that lived near major railyards in San Bernardino County and the information gleaned in this study can only be generalized to similar population groups. However, random sampling was used to recruit participants from each community stratum and as a result, the recruited sample was an ethnically diverse group of community participants from varying educational background and work profiles, including the unemployed and homeless. Also, it is important to note that outcomes were determined and analyses were conducted cross-sectionally, therefore a cause-effect relationship cannot be established.

The focus of this study was only on those who responded to whether they received a doctor diagnosed respiratory illness or heart disease risk factors. No verification of diagnoses was available. QoL and mastery can be helpful as a tool in developing interventions. However, they should not solely be used to assess risk and health outcomes as perceived health status may not measure actual health status. Finally, since the purpose of Project ENRRICH was to characterize the health burden in the residential areas near the San Bernardino Railyard as part of an epidemiological study, there were limitations on what measurement tools were available. A standardized measurement tool for QoL or mastery was not included in the survey. However, the QoL scale was created based on the data from the published qualitative study, Experiences of a railyard community: Life is hard [25] that asked 65 local community members about their perceived QoL and health challenges which included their perception of the potential effects of air pollution on their families. Their responses were organized into thematic topics that included factors such as community violence, social cohesion, walking environment, availability of health foods, perceived control over their lives, and sleep quality. To measure mastery, the researchers for this study utilized a single question, which is identical to one of the 7 items from the Pearlin Mastery Scale (PMS) [26]. In addition, single items have been used to measure constructs [29], such as happiness [30], with good reliability and validity [29,30].

Although the findings are through a study conducted in 2011, the results are still highly relevant, especially in light of the fact that there has been little published to-date on communities living close to major freight railyards. A large community lives in this region (within five miles of the railyard) and is adversely affected by air pollutants. However, we are now only learning the range of health impacts. These health impacts may actually differ from those produced by more vehicular traffic, given that diesel pollutants from major freight railyard include other toxic chemicals (i.e., benzene). The railyard activity has only increased over the years and given the demands for goods across the nation, is expected to increase into the future. There clearly is a critical public health need to highlight the adverse effects that are known and that little has been studied among vulnerable populations living in close proximity to freight railyards. Additionally, there is a need for mitigation efforts reducing exposures or adults and children.

## 5. Conclusions

Although the analysis did not show mastery and QoL as protective buffers against HDRF and RI across all three zones, there were significant relationships between mastery, QoL, and health status at the bivariate level for the zones closest to the railyard. Such findings warrant further investigation, specifically, in light of the potential for disproportionate, compounded, health impacts (for different ethnicities and neighborhoods) experienced as a result of the community environment. In combination with the large body of existing evidence, the results from the current study should be considered relative to their public health implications. In addition, further studies should examine if a change in attitudes toward their health and community lead to altered health status. The researchers suggest that when designing effective policies and programming that would benefit affected communities, one also needs to understand community perceptions about health and well-being including their perceived overall sense of control over one’s life and QoL.

## Figures and Tables

**Table 1 ijerph-15-02765-t001:** Descriptive statistics: categorical variables by zones (*n* = 684).

		Zone 1 (*n* = 210)	Zone 2 (*n* = 228)	Zone 3 (*n* = 246)
		*n* (%)	*n* (%)	*n* (%)
Gender				
	Female	122 (58.1%)	159 (69.7%)	161 (65.4%)
	Male	88 (41.9%)	69 (30.3%)	85 (34.6%)
Ethnicity				
	Latino	163 (77.6%)	152 (66.6%)	162 (65.9%)
	White	16 (7.6%)	26 (11.4%)	46 (18.7%)
	Black	12 (5.7%)	32 (14.0%)	27 (11.0%)
	Other	1 (0.5%)	4 (1.8%)	1 (0.4%)
Marital Status				
	Not married	124 (59.0%)	130 (57.0%)	124 (50.4%)
	Married	86 (41.0%)	98 (43.0%)	122 (49.6%)
Education				
	High School or less	143 (68.1%)	138 (60.5%)	151 (61.4%)
	Some College or More	67 (31.9%)	90 (39.5%)	95 (38.6%)
Income				
	Less than $30,000	159 (75.7%)	146 (64.0%)	150 (61.0%)
	$30,000 or More	51 (24.4%)	82 (36.0%)	96 (39.0%)
Insurance				
	No	101 (48.1%)	88 (38.6%)	94 (38.2%)
	Yes	109 (51.9%)	140 (61.4%)	152 (61.8%)
Health Care Access				
	No	59 (28.1%)	58 (25.4%)	55 (22.4%)
	Yes	151 (71.9%)	170 (74.6%)	191 (77.6%)
HDRF				
	No	149 (71.0%)	162 (71.1%)	188 (76.4%)
	Yes	61 (29.0%)	66 (28.9%)	58 (23.6%)
RI				
	No	166 (79.0%)	165 (72.4%)	188 (76.4%)
	Yes	44 (21.0%)	63 (27.6%)	58 (23.6%)

**Table 2 ijerph-15-02765-t002:** Categorical variables and dependent variable (health status) measured at bivariate level (*n* = 684).

	Zone 1 (*n* = 210)		Zone 2 (*n* = 228)		Zone 3 (*n* = 246)	
	*m* (*sd*)	*t*	*m* (*sd*)	*t*	*m* (*sd*)	*t*
Gender						
Female	3.44 (0.71)	−1.89	3.57 (0.82)	−1.03	3.56 (0.77)	−1.69
Male	3.61 (0.82)		3.67 (0.77)		3.72 (0.76)	
Ethnicity						
Latino	3.46 (0.75)	−2.31 *	3.57 (0.78)	−1.00	3.55 (0.77)	−1.97 *
Non-Latino	3.72 (0.74)		3.67 (0.78)		3.74 (0.80)	
White	3.65 (0.79)	0.78	3.94 (0.62)	3.12 **	3.86 (0.76)	2.54 *
Non White	3.50 (0.76)		3.57 (0.82)		3.56 (0.78)	
Black	3.83 (0.71)	1.89	3.53 (0.92)	−0.66	3.62 (0.82)	0.06
Non-Black	3.49 (0.76)		3.62 (0.82)		3.61 (0.78)	
Marital Status						
Not married	3.59 (0.78)	2.18 *	3.59 (0.85)	−0.25	3.58 (0.78)	−0.58
Married	3.40 (0.71)		3.61 (0.76)		3.63 (0.80)	
Education						
High School or less	3.44 (0.74)	−2.12 *	3.60 (0.81)	−0.9	3.52 (0.83)	−2.44 *
Some College or More	3.63 (0.78)		3.61 (0.81)		3.73 (0.70)	
Income						
Less than $30,000	3.49 (0.76)	−0.67	3.54 (0.83)	−1.7	3.56 (0.81)	−1.56
$30,000 or More	3.54 (0.75)		3.70 (0.76)		3.69 (0.73)	
Insurance						
No	3.52 (0.74)	0.42	3.74 (0.74)	2.60 **	3.64 (0.79)	0.71
Yes	3.49 (0.77)		3.51 (0.84)		3.58 (0.78)	
Health Care Access						
No	3.48 (0.76)	−0.57	3.67 (0.79)	1.58	3.70 (0.73)	1.70
Yes	3.53 (0.76)		3.53 (0.82)		3.55 (0.82)	
HDRF						
No	3.66 (0.72)	4.56 ***	3.74 (0.79)	4.15 ***	3.73 (0.78)	4.13 ***
Yes	3.16 (0.73)		3.26 (0.81)		3.25 (0.75)	
RI						
No	3.56 (0.76)	1.75	3.72 (0.78)	3.93 ***	3.73 (0.74)	4.17 ***
Yes	3.34 (0.76)		3.25 (0.87)		3.24 (0.88)	

Note: * *p* < 0.05; ** *p* < 0.01; *** *p* < 0.001.

**Table 3 ijerph-15-02765-t003:** Correlations between: continuous variables and dependent variable (health status) measured at bivariate level for Zone 1 (*n* = 210).

		*m* (*sd*)	1	2	3	4
1	Health Status (1–5)	3.50 (0.76)	-			
2	Age (19–84)	44.84 (15.73)	−0.15 *	-		
3	Mastery (1–5)	4.15 (0.94)	0.20 *	−0.07	-	
4	QoL (1–5)	3.04 (0.43)	0.20 *	−0.03	0.09	-
5	Functional Impairment (1–4)	4.45 (1.02)	−0.32 **	0.11	−0.18 **	−0.25 **

Note: * *p* < 0.01; ** *p* < 0.001.

**Table 4 ijerph-15-02765-t004:** Correlations between: continuous variables and dependent variable (health status) measured at bivariate level for Zone 2 (*n* = 228).

		*m* (*sd*)	1	2	3	4
1	Health Status (1–5)	3.60 (0.81)	-			
2	Age (18–83)	43.64 (14.00)	−0.18 *	-		
3	Mastery (1–5)	4.31 (0.86)	0.23 **	−0.17 *	-	
4	QoL (1–5)	3.22 (0.46)	0.23 **	−0.02	0.14 *	-
5	Functional Impairment (1–4)	4.29 (1.20)	−0.47 **	0.28 **	−0.30 **	−0.18 *

Note: * *p* < 0.01; ** *p* < 0.001.

**Table 5 ijerph-15-02765-t005:** Correlations between: continuous variables and dependent variable (health status) measured at bivariate level for Zone 3 (*n* = 246).

		*m* (*sd*)	1	2	3	4
1	Health Status (1–5)	3.61 (0.78)	-			
2	Age (20–84)	44.73 (14.10)	−0.09	-		
3	Mastery (1–5)	4.23 (0.87)	0.08	−0.23 **	-	
4	QoL (1–5)	3.31 (0.49)	0.16 *	−0.07	0.23 **	-
5	Functional Impairment (1–4)	4.39 (1.08)	−0.44 **	0.17 *	−0.26 **	−0.31 **

Note: * *p* < 0.01; ** *p* < 0.001.

**Table 6 ijerph-15-02765-t006:** Hierarchal linear regression model predicting health status for Zone 1 (*n* = 196).

Variables	Health Status		
	Step 1 ***	Step 2 ***	Step 3 ***
Age	−0.10	−0.02	−0.02
Latino vs. Non-Latino	0.17*	0.16 *	0.16 *
Marital Status	0.03	0.02	0.01
Education	0.06	0.05	0.03
Functional Impairment	0.30***	0.24 **	0.20 **
HDRF		−0.26 ***	−0.25 **
Mastery			0.03
QoL			0.13
*R^2^*	0.13	0.19	0.20
Adjusted *R^2^*	0.11	0.16	0.17
Δ*R^2^*	0.13	0.06 ***	0.02
*F*-statistic	2.95	7.21	5.91

Note. * *p* < 0.05, ** *p* < 0.01, *** *p* < 0.001.

**Table 7 ijerph-15-02765-t007:** Hierarchal linear regression model predicting health status for Zone 2 (*n* = 216).

Variables	Health Status		
	Step 1 ***	Step 2 ***	Step 3 ***
Age	−0.03	−0.01	−0.01
White vs. Non-White	−0.13 *	−0.14 *	−0.15 *
Insurance	−0.14 *	−0.14 *	−0.14 *
Functional Impairment	0.49 ***	0.42 ***	0.36 ***
HDRF		−0.10	−0.09
RI		-.12 *	−0.10
Mastery			0.14 *
QoL			0.12 *
*R^2^*	0.30	0.32	0.35
Adjusted *R^2^*	0.28	0.30	0.31
Δ*R^2^*	0.30	0.02 *	0.03 **
*F*-statistic	21.77	10.75	9.83

Note. * *p* < 0.05, ** *p* < 0.01, *** *p* < 0.001.

**Table 8 ijerph-15-02765-t008:** Hierarchal linear regression model predicting health status for Zone 3 (*n* = 237).

Variables	Health Status		
	Step 1 ***	Step 2 ***	Step 3 ***
Age	−0.04	0.00	0.01
Latino vs. Non Latino	0.14	0.15 *	0.16 *
White vs. Non-White	−0.13	−0.12	−0.13
Education	0.08	0.08	0.06
Functional Impairment	0.46 ***	0.40 ***	0.36 ***
HDRF		−012	−0.12
RI		−0.14 *	−0.14 *
QoL			0.11
*R^2^*	0.25	0.29	0.30
Adjusted *R^2^*	0.23	0.26	0.27
Δ*R^2^*	0.25	0.04 **	0.02
*F*-statistic	15.01	13.06	10.8

Note. * *p* < 0.05, ** *p* < 0.01, *** *p* < 0.001.

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
