# Peer review of "Impact of Heart Disease Risk Factors, Respiratory Illness, Mastery, and Quality of Life on the Health Status of Individuals Living Near a Major Railyard in Southern California"

_ijerph, 2018, doi:10.3390/ijerph15122765_

Round 1

Reviewer 1 Report

Why ‘Major Transportatin Hub’? This may elicit connotations associated with multi-modal transportation, such as bus rapid transit or light rails. Why not just say railyard?

 Your title suggests that this piece is about freight gateways as an environmental health issue. However, the abstract and conclusion don’t quite help readers connect what this study has to do with railyards. Also, are you linking the constructs of optimism and Q of L to the railyard and related built environment? You get at this in your discussion section a bit but help readers see this as part of your hypotheses— if it is.

 Maybe cite resources on CBPR or very briefly define it. Even though most people know what it is, it could be helpful to redirect readers that do not.

Regarding methods, say more. Can you explain the random selection process? How were households within the zones selected? Why were these three zone distances selected? How long were the interviews? Who conducted them?  What were the respiratory tests? How did you ensure standard protocol were used? It’s unclear if you created the Q of L and optimism scales or used an existing psychometric tool.  

In Table 1, explain to readers what the p-value represents so it can stand alone.

Regarding limitations, say more. You mention the limitations of only capturing diagnosed heart disease and respiratory illness. Both are underdiagnosed, especially in under-insured populations. Did you ask about other related respiratory issues, in general? Why or why not?

The data are now 7 years old. Help readers to see why this is still worth publishing or acknowledge in limitations if there are any implications of these older data.

Author Response

Reviewer 1 Comments

Point 1: Why ‘Major Transportation Hub’? This may elicit connotations associated with multi-modal transportation, such as bus rapid transit or light rails. Why not just say railyard?

Response: The term “major transportation hub” was replaced with  “railyard” in the manuscript.

 Point 2: Your title suggests that this piece is about freight gateways as an environmental health issue. However, the abstract and conclusion don’t quite help readers connect what this study has to do with railyards. Also, are you linking the constructs of optimism and Q of L to the railyard and related built environment? You get at this in your discussion section a bit but help readers see this as part of your hypotheses— if it is.

Response: It was clarified that the premise of this study was to see what types of potential resources this very marginalized community had access and the emphasis was that in addition to the socio-economic challenges, this community also had to face environmental factors that had significantly negative effects on their health.

Point 3: Maybe cite resources on CBPR or very briefly define it. Even though most people know what it is, it could be helpful to redirect readers that do not.

Response: Included brief description of community based participatory research (CBPR) and clarified the use of CBPR in the Project ENRICH study.

Point 4: Regarding methods, say more. Can you explain the random selection process? How were households within the zones selected? Why were these three zone distances selected? How long were the interviews? Who conducted them?  What were the respiratory tests? How did you ensure standard protocol were used? It’s unclear if you created the Q of L and optimism scales or used an existing psychometric tool.  

Response: Further descriptions were added to the methods section. Specifically, is was added that zone distances were selected based on the prevalence of adverse health effects among exposed adults. The zones were defined across the spatial gradient of air pollution and associated health risks based on the 2011 California Air Resources Board Health Risk Assessment report that was originally derived through the implementation of computer-based modeling used for estimating the transport and dispersion of diesel emissions from the railyard. The sampling strategy included every household in Zone 1 (less than a mile from the railyard) was sampled to match the sharp decline of diesel particulate matter concentrations which was postulated to occur over a short distance from the railyard. Sampling in zones 2 and 3 employed a t-stage clustering sampling methodology in which sixty census blocks within in sampling zone were first randomly selected and then a set of fie houses within each Census block was chosen. Selected houses were randomly selected using a GIS-based random number generator. The data were collected by trained research assistants with interviews, in English or Spanish based on participant’s preference, lasting for approximately 60-90 minutes. Non-invasive biological tests were also conducted (respiratory tests such as using NIOX MINO devices to collect airway inflammation data via fractional exhaled nitric oxide) in addition to collecting information on demographics, exposures, health status, lifestyle, self and community perceptions.

Additional details of how the Quality of Life (QoL) scale was developed was added to the QoL scale section. The Quality of Life scale was created based on the data from the published qualitative study, Experiences of a railyard community: Life is hard by Spencer-Hwang, Montgomery, Dougherty, Valladares, Gleason, & Soret (2014) that asked 12 local community members about their perceived quality of life and health challenges which included their perception of the potential effects of air pollution on their families. Their responses were organized into thematic topics that included factors such as community violence, social cohesion, walking environment, availability of health foods, perceived control over their lives, and sleep quality.  See below for the full citiation

Spencer-Hwang, R., Montgomery, S. Dougherty, M., Valladares, J., Rangel, S.,…Soret, S. (2014). Experiences of a rail yard community: Life is hard. Journal of Environmental Health, 77(2), 8-17.

After further review and analysis, the “optimism” construct has been re-named “mastery”. This question has identical to one of the 7 items from the Pearlin Mastery Scale (PMS) (1978) that measures the “extent to which an individual regards their life chances as being under their personal control rather than fatalistically ruled.” The survey also included “I often feel helpless in dealing with the problems in life” which is also an item in the PMS. However, the two items were not combined to create the variable due to the poor internal reliability (Cronbach’s α = 0.29). These were the only two items that appeared to measure mastery in the survey and due to the poor internal reliability between the items, the item with the higher correlation to health status was selected. Please see below for the full citation.

Pearlin, L. I., & Schooler, C. (1978). The structure of coping. Journal of Health and Social Behavior, 2-21.

Point 5: In Table 1, explain to readers what the p-value represents so it can stand alone.

Response: Deleted the p-value column in table. Note at the bottom of the table to explain level of significance.  

Point 6: Regarding limitations, say more. You mention the limitations of only capturing diagnosed heart disease and respiratory illness. Both are underdiagnosed, especially in under-insured populations. Did you ask about other related respiratory issues, in general? Why or why not?

Response: Study did ask about other respiratory illnesses (RI) and heart disease risk factors (HDRI). The results of RI issues is cited/reported elsewhere. There were several different HD variables. However, to make the model parsimonious, we chose the most commonly reported HDRI variables that were above the state and national benchmark to create the HDRI and RI variable.

Point 7: The data are now 7 years old. Help readers to see why this is still worth publishing or acknowledge in limitations if there are any implications of these older data.

Response: The findings through a study conducted 7 years ago are still highly relevant, especially in light of the fact that there has been little published to-date on communities living in close to major freight railyards. A large community lives in this region (within five miles of the railyard) and adversely affected by air pollutants and we are now only learning the range of health impacts.  These health impacts may actually differ from those produced by more vehicular traffic, given that diesel pollutants from major freight railyard include other toxic chemicals (i.e. benzene). The railyard activity has only increased over the years and given the demands for goods across the nation, is expected to increase into the future. There clearly is a critical public health need to highlight the adverse effects that are known and that little has been studied among vulnerable populations living in close proximity to freight railyards. Additionally, there is a need for mitigation efforts reducing exposures or adults and children.

Reviewer 2 Report

see attached file 

Author Response

Reviewer 2 Comments

Point 1: The guiding hypothesis of this study appears to be that individuals living nearer to a major transportation hub would have more compromised quality of life and other health status measures. To their credit the investigators conducted extensive interviews of a random household sample including 670 residents who are predominantly Latino. They sampled individuals who lived in three different strata based on distance from rail hub but the distances are relatively short with a maximum of only 5 miles from the hub. Not surprisingly they do not find a significant association with distance from the hub with their outcomes. But they do not have an adequate control group to claim that health status or other factors are not affected by this exposure.

Response: We know that the poorest, least educated live closest to the railyard so and that distance is an issue. Therefore to address this comment, the data were separated by zones and the bivariate and multivariate analyses were re-run to more accurately assess if health status was impacted by distance to the railyard. Three separate regressions were run and each zone was compared to each other. Results showed that that lived the closest to the railyard were most negatively impacted by heart disease risk factors (HDRI) without the possible protection of optimism and/or perceived quality of life on perceived health status. Respondents that lived further from the railyard were less impacted by HDRI and respiratory illness (RI) in addition to having optimism and quality of life potentially acting as a protective buffer. This helps provide support that further distance from the railyard increases the positive impact of higher perceived quality of life and optimism on better health status.

Point 2: Another hypothesis was that optimism and quality of life would be associated with improved self- reported health status. This hypothesis does not appear to be related to the railyard exposure, other than the individuals in this analysis live in more challenging circumstances. A major concern is their failure to offer any references for their measures of quality of life and optimism to support validity and reliability. Another significant measure is healthcare access which also does not have any reference.

Response: The focus on the study was to assess if “optimism” (which as been re-defined as “mastery”) and quality of life (QoL) acted as a potential protective buffer for heart disease risk factors (HDRI) and/or respiratory illness in a highly stressed, vulnerable community that also had to cope with the additional burden of environmental pollutants due to living near a major railyard.

Additional details of how the Quality of Life (QoL) scale was developed was added to the QoL scale section. The Quality of Life scale was created based on the data from the published qualitative study, Experiences of a railyard community: Life is hard by Spencer-Hwang, Montgomery, Dougherty, Valladares, Gleason, & Soret (2014) that asked 12 local community members about their perceived quality of life and health challenges which included their perception of the potential effects of air pollution on their families. Their responses were organized into thematic topics that included factors such as community violence, social cohesion, walking environment, availability of health foods, perceived control over their lives, and sleep quality.  See below for the full citation.

Spencer-Hwang, R., Montgomery, S. Dougherty, M., Valladares, J., Rangel, S.,…Soret, S. (2014). Experiences of a rail yard community: Life is hard. Journal of Environmental Health, 77(2), 8-17.

After further review and analysis, the “optimism” construct has been re-named “mastery”. This question has identical to one of the 7 items from the Pearlin Mastery Scale (PMS) (1978) that measures the “extent to which an individual regards their life chances as being under their personal control rather than fatalistically ruled.” The survey also included “I often feel helpless in dealing with the problems in life” which is also an item in the PMS. However, the two items were not combined to create the variable due to the poor internal reliability (Cronbach’s α = 0.29). These were the only two items that appeared to measure mastery in the survey and due to the poor internal reliability between the items, the item with the higher correlation to health status was selected. Please see below for the full citation.

Pearlin, L. I., & Schooler, C. (1978). The structure of coping. Journal of Health and Social Behavior, 2-21.

The healthcare access was not based on a standardized measure. Respondents were asked if there was a place that the respondents went to when sick or in need of advice about their health. If they replied yes, they were probed on where they went, for example a physician’s office, clinic, county public health department clinic, or other specified locations. Those who did not have access or stated that they went to the emergency room were categorized as not having regular health care access. The responses were collapsed and then dichotomized with ‘0’ indicating that they did not have access to regular care and ‘1’ indicating that they did have access to regular care. These are general questions regarding access to health care that is not exclusive to a measurement tool or a specific study. Therefore, we do not believe that it would warrant a citation. 

Point 3: As in many other studies they find that health status varies by race/ethnicity but they do not find any variation by proximity to the railyard. They also found that more optimistic respondents had better health status which seems quite plausible but not particularly novel. In regard to analysis of the influence of heart disease – it is a little disturbing when they combine coronary artery disease with just having hypertension; the severity of heart disease varies widely which probably undermines their analysis related to quality of life and optimism.

Response: As stated in response to the first comment, we know that the poorest, least educated live closest to the railyard so and that distance is an issue. Therefore to address this comment, the data were separated by zones and the bivariate and multivariate analyses were re-run to more accurately assess if health status was impacted by distance to the railyard. Three separate regressions were run and each zone was compared to each other. Results showed that that lived the closest to the railyard were most negatively impacted by heart disease risk factors (HDRI) without the possible protection of optimism and/or perceived quality of life on perceived health status. Respondents that lived further from the railyard were less impacted by HDRI and respiratory illness (RI) in addition to having optimism and quality of life potentially acting as a protective buffer. This helps provide support that further distance from the railyard increases the positive impact of higher perceived quality of life and optimism on better health status.

Point 4: The discussion tries to explain why they do not find any differences in regard to exposure to air pollutants because but they really do not have an adequate control group. They expend some effort trying to explain this finding on page 7 lines 253 through 257 but do not adequately address the limitations of their study.

Response: Again to address comments regarding not having an adequate control group in regard to exposure to air pollutants, the data were separated by zones and the bivariate and multivariate analyses were re-run to more accurately assess if health status was impacted by distance to the railyard. Three separate regressions were run and each zone was compared to each other. Results showed that that lived the closest to the railyard were most negatively impacted by heart disease risk factors (HDRI) without the possible protection of optimism and/or perceived quality of life on perceived health status. Respondents that lived further from the railyard were less impacted by HDRI and respiratory illness (RI) in addition to having optimism and quality of life potentially acting as a protective buffer. This helps provide support that further distance from the railyard increases the positive impact of higher perceived quality of life and optimism on better health status.

Point 5: The references that they offer to justify the effect of air pollution on health status are out of date and there are much more useful studies. They should also acknowledge that the main pollutant of concern is diesel according to the studies that were done in the region of that railyard in the past. The concern is that now there are so many highways that crisscross this region that it is hard again to find a group that is not exposed.

Response: References to justify the effect of air pollution on health status has been updated. We thank the reviewer for the suggested references listed below which have been included in the manuscript.

Reviewer 3 Report

I enjoy reading your paper, it is well written, however, I suggest you address the following:

Abstract should mention specific statistical analyses conducted

In addition describe your main findings with data in the abstract

Page 3, line 123: state the IRB body the reviewed the study not just the university

Page 3 line 140: define COPD at first use

Page 4 line 179: 'majority' ?(66.4%) maybe 'two-thirds"

Since you have a couple of variables in your model, did you assess any possible variable interactions that could be ending in multicollinearity? This can significantly  impact your findings 

Author Response

Reviewer 3 Comments

Point 1: Abstract should mention specific statistical analyses conducted in addition describe your main findings with data in the abstract

Response: It was added that three separate hierarchal linear regressions were run to the abstract.

Point 2: Page 3, line 123: state the IRB body the reviewed the study not just the university

Response: Changed to approved by the university and Institutional Review Board

Point 3: Page 3 line 140: define COPD at first use

Response: Defined COPD as chronic obstructive pulmonary disease followed by the acronym (COPD)

Point 4: Page 4 line 179: 'majority' ?(66.4%) maybe 'two-thirds"

Response: Changed majority to two-thirds

Point 5: Since you have a couple of variables in your model, did you assess any possible variable interactions that could be ending in multicollinearity? This can significantly  impact your findings 

Response: The interactions between the illness variables (heart disease risk factors and respiratory illness) and the protective buffers (optimism and quality of life) were omitted from the final model due to VIF scores indicating that there were multicollinearity (outside of the acceptable range of -2 to 2). All other interactions between the variables were within the normal range. 

Round 2

Reviewer 1 Report

Nice job addressing missing methodological details. The manuscript is much stronger. Some additional feedback:

- There are some minor writing issues remaining (e.g., in abstract - this variable vs. these variables, add in (QoL); fie houses).

 Some of the info under ‘The Enrich Project’ header seems like it should be in methods. At first, I think as though you are telling me about past work they did since this is the introduction.

- I see the relevance of ‘mastery’ but this isn’t a common construct in public health. You may want to introduce it in the intro a bit so readers can understand how it relates better to quality of life and why it’s in your model. Is it like self-efficacy – in that mastery depends on the behavior or issue in context? I’d worry about using a single item as such a central focus without more explanation and justification. Earlier on help us understand how it relates to sense of control (mentioned in the discussion) or does it?

- To previous feedback, the authors explained: “Study did ask about other respiratory illnesses (RI) and heart disease risk factors (HDRI). The results of RI issues is cited/reported elsewhere. There were several different HD variables. However, to make the model parsimonious, we chose the most commonly reported HDRI variables that were above the state and national benchmark to create the HDRI and RI variable.” This justification would be useful to readers to hear…not just reviewers.

Author Response

Point 1: There are some minor writing issues remaining (e.g., in abstract - this variable vs. these variables, add in (QoL); fie houses).

Response: The manuscript was reviewed to address any inconsistencies and writing errors.

Point 2: Some of the info under ‘The Enrich Project’ header seems like it should be in methods. At first, I think as though you are telling me about past work they did since this is the introduction.

Response: The details of the Project ENRICH study were included in the introduction to provide background information on the uniqueness of the population, relevance, and justification for the study. This information was included in introduction (in addition to the methods section) to also provide context as to why the analyses were separated by zones. While it may appear to be redundant, the information in the methods section also goes into further details about the sampling method and how the zones were created.  

Point 3: I see the relevance of ‘mastery’ but this isn’t a common construct in public health. You may want to introduce it in the intro a bit so readers can understand how it relates better to quality of life and why it’s in your model. Is it like self-efficacy – in that mastery depends on the behavior or issue in context? I’d worry about using a single item as such a central focus without more explanation and justification. Earlier on help us understand how it relates to sense of control (mentioned in the discussion) or does it?

Response: Background information about the concept ‘mastery’ and the heart disease risk factors, respiratory illness, and health status were strengthened in the introduction on page 4 (please refer to the highlighted area).

Point 4: To previous feedback, the authors explained: “Study did ask about other respiratory illnesses (RI) and heart disease risk factors (HDRI). The results of RI issues is cited/reported elsewhere. There were several different HD variables. However, to make the model parsimonious, we chose the most commonly reported HDRI variables that were above the state and national benchmark to create the HDRI and RI variable.” This justification would be useful to readers to hear…not just reviewers.

Response: Further detail in regard to rationale for the development of HDRI and RI variable was added to the methods section on page 5 (please refer to the highlighted area).